# Interventions on Food Security and Water Uses for Improving Nutritional Status of Pregnant Women and Children Younger Than Five Years in Low-Middle Income Countries: A Systematic Review

**DOI:** 10.3390/ijerph18094799

**Published:** 2021-05-02

**Authors:** Cristina Urgell-Lahuerta, Elena Carrillo-Álvarez, Blanca Salinas-Roca

**Affiliations:** 1Department of Nursing and Physiotherapy, University of Lleida, Montserrat Roig 2, 25198 Lleida, Spain; cristinaurgells12@gmail.com; 2Global Research on Wellbeing (GRoW) Research Group, Blanquerna School of Health Science, Ramon Llull University, Padilla 326-332, 08025 Barcelona, Spain; elenaca@blanquerna.url.edu

**Keywords:** malnutrition, food insecurity, pregnancy, children, WASH

## Abstract

Malnutrition is a global health issue concerning children and pregnant women in low- and middle-income countries (LMICs). The aim of this review was to assess the health-impact outcomes of interventions addressing food security, water quality and hygiene in order to address the improvement of the nutritional status in children below five years and pregnant women in LMICs. Using PRISMA procedures, a systematic review was conducted by searching in biomedical databases clinical trials and interventions for children and pregnant women. Full articles were screened (n_f_ = 252) and critically appraised. The review included 27 randomized and non-randomized trials and interventions. Based on the analysis, three agents concerning nutritional status were identified. First, exclusive breastfeeding and complementary feeding were fundamental elements in preventing malnutrition. Second, provision of sanitation facilities and the promotion of hygienic practices were also essential to prevent infections spread and the consequent deterioration of nutritional status. Finally, seasonality was also seen to be a relevant factor to consider while planning and implementing interventions in the populations under study. In spite of the efforts conducted over last decades, the improvement in food insecurity rates has remained insufficient. Therefore, the development of global health programs is fundamental to guide future actions.

## 1. Introduction

In the past, proper nutrition, safe water, hygiene and sanitation have been identified as priorities for ensuring global health. Malnutrition is a complex paradigm that mainly affects unprivileged populations, where basic requirements such as a balanced diet, hygiene, sanitation and healthcare are limited. This is linked with food insecurity, widely-spread infection and disease, and water use; it is also tied to climate change, since dry and rainy seasons affects food crops and contamination of water, especially in tropical climates, thus affecting most low- and middle-income countries (LMIC). In order to approach the challenge of malnutrition, different international and local interventions have been performed to address food security (FS) and the water uses of sanitation and hygiene (WASH). Furthermore, the 2030 Agenda for Sustainable Development Goals (SDG) adopted by the United Nations in 2015 identifies the importance of ending hunger and malnutrition (SDG #2), clear water, and sanitation (SDG #6) for global health.

Concerning FS, four key ambits are defined: availability (adequate quantity of food), access (food resources available), utilisation (nutritious and safe diets as well as clean water) and stability (the temporal aspect of the other three dimensions). Nowadays, for many countries, FS remains a critical challenge [1,2], and the situation is worsening in most regions in LMICs. The State of Food Security and Nutrition in the World 2020 described contributing factors such as adverse climate events, instability in conflict-ridden regions and economic slowdowns. Concretely, is estimated that at least 340 million children suffer from micronutrient deficiencies, 144 million children under the age of five years are too short for their age, 47 million children under the age of five years are thin for their height, and 790 million people have insufficient daily dietary energy intake [3]. Moreover, actual evidence has shown that dietary diversity factors can reverse this situation; however, future studies are required to understand diverse and dynamic foods systems in the context of climate change [4]. 

In addition, the role of WASH is a key factor for addressing malnutrition in terms of the control of enteric infections and the provision of food security and water supply and sanitation, particularly in LMICs [5]. Moreover, precipitation extremes (including increased rainfall and prolonged drought) lead to a high exposure to pathogenic bacteria, parasites, toxins and viruses, which affect water use. The result of the exposure to these pathogens is a huge range of infections and diseases, which consequently have a great impact on the nutritional status, growth and development of people affected [6]. Additionally, the presence of safe water and sanitation systems should be considered as a critical point for FS. However, access to potable water supplies and proper sanitation facilities still eludes a large proportion of LMICs. For that reason, the applicability of such interventions is very wide and can be generally classified in four different main categories: water sources (i.e., piped water systems), water quality (i.e., water filtration and treatment systems), sanitation (i.e., toilets, latrines) and education (i.e., practice of hand washing). The benefits of such interventions are substantive and highly cost-effective at macroeconomic as well as household levels, according to the World Bank. In fact, the growth per capita (GDP) of poor countries with improved access to water and sanitation is much higher than that of equally poor countries without improved access [7,8]. Apart from the benefits on the reduction of disease and mortality, sanitation and access to water supply also boost school attendance and reduce dropout rates. 

Currently, children below five years are those more vulnerable to suffering health impacts caused by food insecurity. Likewise, the nutritional status of pregnant women seems to be crucial to further child growth. Regarding pregnant women, 32 million are anaemic and 19 million suffer from vitamin A deficiency. Since there has been evidence quantifying and describing malnutrition in pregnant women and children below five years, interventions with regards to the nutritional status of that vulnerable population have been performed. Analysing the influence of such interventions performed in the field of FS and WASH with regard to children below five years and pregnant women in LMICs is a key aspect in global health. Nevertheless, there are no studies addressing the nexus between FS, WASH, climate change and malnutrition in pregnant women and children below five years.

In this sense, the present systematic review attempts to assess the interventions performed on FS and WASH, considering the nutritional status of pregnant women and children below five years in LMICs, in order to describe the nexus of malnutrition in the current context of climate change as an approach to elucidate further needed interventions.

## 2. Materials and Methods

### 2.1. Design

A systematic review of argument-based literature in order to better understand the meanings, foundations and impact of interventions on FS and WASH with regard to nutritional status was carried out following the PRISMA methodology. First, we formulated our research questions; second, we conducted a systematic literature search using a combination of keywords as described in PRISMA methodology [9]; third, we identified and described the different relations among FS and WASH interventions and the impact on the nutritional status of pregnant women and children below five years.

### 2.2. Research Question

To the best of our knowledge, no published reviews exist that specifically focus on the impact of FS and WASH interventions on the nutritional status of pregnant women and children below five years. This prompted us to formulate the following research question: What is the impact of water quality and food security interventions on the nutritional status of pregnant women and children below five years in LMICs?

### 2.3. Literature Search and Selection Criteria

Studies that examined the association between food insecurity and WASH in pregnant women and children in LMICs were identified in journals indexed by PubMed and Scopus. The search was performed in databases (PubMed and Scopus) that would be able to provide a wider range of article type. The search was performed by means of the combination of keywords and Boolean operators (e.g., AND and OR). The search concluded in May 2020. For the systematic review, no grey literature was used. The keywords included were “food security”, “nutritional status”, “low- and middle-income countries”, “water, sanitation”, “hygiene WASH”, “malnutrition”, “water contaminants” and “intervention”, linked by Boolean operators. We also applied the filter “pregnancy”, “pregnant”, “undernutrition” and “children”. The search string is described in Table 1.

We included the studies that described the interventions and effects in FS or WASH on pregnant women or children below five years. Case studies of individuals, letters and editorials were not eligible. We considered studies published from 2000 to 2020.

### 2.4. Search Outcomes

We focused on the effects of household food insecurity levels, children’s anthropometric measurements, maternal education and wealth index.

### 2.5. Quality Appraisal

The quality of the approved studies was assessed using the Critical Appraisal Skill Program (CASP) checklist. This checklist comprises 12 questions, with a score of 0 or 1. Articles with a CASP score equal to or higher than 9/12 were considered suitable.

## 3. Results

In this section we report the characteristics and results of the articles in the final sample. Three groups of intervention are described: (i) food-based, (ii) food-based with WASH interventions and (iii) WASH (see Table 2). Most original studies considered several of the listed outcome variables and were generally measured in the same units in individual studies.

Regarding the variable of household food insecurity levels, data was recorded according the Household Food Insecurity Access Scale (HFAIA) and subsequently classified in four different categories (food secure, mildly food insecure, moderate food insecure, severe food insecure).

In addition, the measurement of children’s anthropometry generally included variables such as weight and height.

### 3.1. Study Selection

The initial search identified 3290 titles (n_t_). Reasons for exclusion were considered for at least one of the following items: articles not fitting the aim of the review; articles with no description of the collection system used; articles with only a biochemical and not a clinical focus; and/or papers written before 2000. As a result, 252 articles (n_f_) were kept after applying filters, and 3038 were excluded. Duplicates were removed, and 233 articles remained for further eligibility criteria. Afterwards, articles were filtered by title or abstract, and 57 full-text articles were kept for a further eligibility assessment. Such assessment consisted of reading full-text articles, to establish whether the article could be relevant to qualitative synthesis. Finally, 27 articles (N) were included to develop the systematic review according to the eligibility criteria (Figure 1).

### 3.2. Study Characteristics

Our dataset includes an assortment of interventions conducted in several LMICs taken from 27 original studies: 16 randomized-control trials, six cross-sectional studies, three prospective longitudinal trials, one differences-in-differences analysis and one study protocol (Table 2). 

Such interventions were based on clinical measures, household hygienic improvements, and nutrition and WASH counselling, focused on children below five years and pregnant women. The nutritional improvements varied as a result of the promotion of breastfeeding and appropriate complementary feeding, the provision of nutritional supplementation in combination with maternal education, and the promotion of hygienic practices such as hand washing.

### 3.3. Food-Based and Nutritional Interventions 

Among the food-based and nutritional approaches to food insecurity, nutritional education counselling (CEN) and micronutrient supplementation (MiS) interventions were the most frequent.

Personalized home-based CEN by community health workers on maternal, infant and young child nutrition practices was effective, improving morbidity and nutritional outcomes of infants. Basic nutritional training and/or provision of information materials was significantly adequate in improving exclusive breastfeeding rates in Kenyan communities [10]. Other trials examined the effect of household FS status and its interaction with household wealth status in stunting among children between 6 months and 23 months in resource-poor urban settings in Kenya. Stunting was highest among poorest households and lowest among the least poor. In addition, the proportion of children who were stunted was higher among mothers with lower levels of education [11]. In fact, it was observed that children of mothers with secondary or higher education had a lower risk of childhood stunting, underweight and wasting in Bangladesh [12]. A trial consisting of the delivery of nutrition interventions integrated into an existing neonatal and child health program reduced household FS. Furthermore, high socioeconomic status and maternal education were associated with increased odds of minimum dietary diversity (MDD), whereas household FS was not associated with MDD [13]. 

Another significant consideration in food-based interventions was the dietary diversity among children and pregnant women under food insecurity. Studies determined that women had higher dietary diversity in autumn and winter. Household food insecurity concerns peaked during autumn and were lesser in spring [14]. In India, a study protocol aimed to evaluate the potential of indigenous foods in contributing to dietary diversity and nutrient intake for improving food security and nutritional status of vulnerable groups; this study provided the first comprehensive examination of the food system of tribal communities in India. The study’s aim was to identify interventions to help support the sustainable production and consumption of indigenous foods and address the burden of malnutrition in the tribal communities [4]. In fact, a trial that comprised four LMICs (Guatemala, India, Pakistan and Democratic Republic of Congo) examined the dietary diversity of women’s diets, estimating the usual group energy and nutrient intake during the first trimester of pregnant women participating in the trial. Significantly higher intakes of most key nutrients were observed in participants with adequate dietary diversity. The authors suggested there is a likely need for micronutrient supplementation in pregnancy and support for the value of increasing dietary diversity [15].

Regarding MiS, the evaluation by Shaheen et al. determined that an early intervention in prenatal food supplementation and multiple MiS lowered mortality in children before the age of five years and reduced the gap in child survival chances between social groups. In this study pregnant women were randomized to groups receiving food supplementation during pregnancy or one of three types of micronutrient capsules: 30 mg iron and 400 μmg folic acid, 60 mg iron and 400 μmg folic acid, or multiple micronutrients [16]. In addition, the use of mineral- and vitamin-enhanced micronutrient powders reduced stunting significantly in a group of children younger than six months in Bangladesh [17]. Similarly, in Senegal the delivery of iron-fortified yoghurts through a dairy value chain program meant an increase in the haemoglobin of the intervention group after one year. Anaemia prevalence was very high at baseline (80%) and dropped close to 60% at the endpoint, with no difference between intervention groups. Also, a greater impact was found for boys compared to girls [18].

Locally produced ready-to-use therapeutic food (RUTF) was also proven to be effective in preventing growth faltering and improving micronutrient status [19,20]. Nevertheless, according to Sigh et al., the provision of local foods combined with CEN did not improve young child nutritional status as compared to CEN alone. A few articles studied the impact of using vouchers as a strategy to reduce waste during seasons when food insecurity is at its highest [18,21].

### 3.4. WASH Interventions

Water, sanitation and hygiene (WASH) programs have also obtained encouraging results. In Kenya, the application of a WASH-focused program reported great improvements from baseline, but overall levels of latrine coverage were still low. This suggests that the findings challenge the assumption that providing WASH infrastructure and education will result in behaviour change [22]. However, Nurul et al. obtained a modest behaviour change after a large-scale sanitation, hygiene and water improvement program on childhood diarrhoea and respiratory disease in rural areas. Other studies indicated no reduction in the prevalence of diarrhoea or respiratory disease in children under three years [23]. Focusing deeply on diarrhoea prevalence, Tsuka et al. evaluated the risk factors for undernutrition in urban slums in Indonesia. The results suggested that not using a towel for hand washing practices was significantly associated with an increased risk of stunting. Children from households using tap water as drinking water were associated with an increased risk of stunting and thinness compared with households using tank water. In addition, children from households using open containers for water storage were associated with an increased risk of diarrhoea [24]. In Bangladesh slums, 83% of the selected households experienced food insecurity regarding the microbial quality of food and water consumed by children in the selected slums. Also, mothers or caregivers in slums were less likely to have higher levels of education [25]. The poor personal hygiene practices of mothers, especially hand washing practices before food preparation, may have been a contributing factor to the high contamination rates, as all the collected water samples were contaminated. There was no impact of the community-based WASH intervention in terms of reducing *G. Duodenalis* prevalence. Risk factors for the infection included living in a household with a child under five years of age, living in a household with more than six people, and sampling during the rainy season [26].

### 3.5. Food-Based Interventions Combined with WASH Interventions

The articles that emphasized the importance of addressing stunting using interventions that combine WASH and food insecurity determined that the education to apply chlorine treatment for different water uses (drinking, food washing, etc.) improved sanitation, limiting exposure to faeces and hand washing with soap [22]. Although these are promising results, further research is needed in this area.

## 4. Discussion

This is the first systematic review that considers food-based and WASH interventions in the approach to food security (FS) in LMICs. Persisting global food insecurity is a highly debated issue, with different policies used to enact novel strategies to reduce its impact on global health.

The interventions in this review described the importance of FS and WASH approaches in order to improve nutritional status in young children but also in pregnant women, as crucial to children’s future growth. On one hand, the importance of exclusive breastfeeding (EBF) and complementary feeding practices, including supplementation in many cases, have been demonstrated [27,28]. On the other hand, the effectiveness of improving household sanitation facilities and the promotion of hygienic practices have been shown to be important, including the impact of seasonality on household food insecurity and dietary diversity.

Many studies have shown a decrease in infectious diseases and malnutrition in children when various educational programs are conducted. For this reason, focusing on the reduction of infection prevalence has been a key point to take in account while developing interventions to improve malnutrition rates. In addition, it is important to point out the fact that such interventions were mostly carried out by trained community health workers (CHWs); this apparently contributed to a better acceptance of the interventions and subsequently to obtaining better results. Therefore, owing to the strict influence of food security and WASH on nutritional status, many programs have focused on combined interventions to address malnutrition.

### 4.1. Socioeconomic Variables and Nutritional Status

In addition to food-based and WASH interventions, sociocultural issues affecting food insecurity, such as wealth and maternal education, must be analysed. Women from food-insecure households are less educated than women living in households classified as food secure. Therefore, children growing up with women in food-insecure households might present earlier malnutrition than those living in food-secure households [25]. In addition, wealth index is inversely proportional to the severity of household food insecurity, along with lower dietary diversity and the proportion of stunted children [29]. Consequently, inadequate macronutrient intake is more common among women with lower education, with low socioeconomic status and from food insecure households [30]. Conversely, adherence to prenatal food supplements is higher among women who have less schooling. This may be due to the need of supplementary food as a consequence of the frequent problem of food insecurity [16].

Despite the close relationship between food security, poverty and nutritional status, the mechanism of transmission remains unclear. However, there is no hesitation that FS is a public health problem [11]. Acknowledging that higher levels of maternal education are beneficial for better nutritional outcomes, promoting women’s education at least up to a secondary level is a step to tackle malnutrition [12]. Moreover, educational benefits for child health are independent of household economic means [13].

In an attempt to prevent the burden of disease, cash-based transfer programs are an emerging strategy in the prevention of malnutrition in children. However, the current evidence surrounding the use of either cash or voucher transfer programs is elusive. The Refani Pakistan study provides robust evidence to help increase the understanding of how and why certain modalities of cash transfer work better than others [21]. The multi-billion dollar initiative by the US government—Feed the Future—has been a key point in reducing stunting among children. This initiative has focused efforts on reducing undernutrition and poverty by focusing on nutrition, household income, food security, and agriculture, achieving an important reduction in annual percentages of stunting [31].

### 4.2. Diet Diversity and Seasonality: Factors for Zero Hunger, Considering the Climate Change Context

Dietary diversity and household FS are reported to also be sensitive to seasonal variations. Seasonality is significantly associated with the dietary diversity and FS status of pregnant women. In Bangladesh, pregnant women have higher dietary diversity in autumn and winter. Nevertheless, despite this seasonal fluctuation of dietary diversity, in this trial, a relationship was not identified between seasonality and maternal nutritional status [14].

While modern agricultural practices to improve food sufficiency have increased agricultural productivity, they have also put a strain on the environment. In spite of the increase in the productivity of food crops, the number of people around the world who are food and nutrition insecure remains high. As the low productivity and instability of production are often attributed to the climatic changes in disadvantaged populations, future studies to understand the diverse and dynamic food systems in relation to the paradigm of the climate emergency are fundamental [4].

According to the World Health Assembly 2012 global nutrition targets, a 40% reduction in the number of children younger than five who are stunted should be accomplished by 2025. Many intervention programs have been conducted over the past decades; however, the current trends of many LMICs are still insufficient to reach the target [32]. Further research and investment may contribute to the development of more effective interventions to address this burden of disease.

## 5. Conclusions

This review is the first of its kind to analyse the role of FS and WASH interventions in the nutritional status of pregnant women and children below five years in LMICs. Several limitations within the studies reviewed have been identified. The variation of results regarding the studied country as well as poor response rates and unreported data were apparent in several studies.

The findings of this review suggest that combined food-based and WASH interventions have been largely effective in addressing malnutrition, especially in children. Results show the outstanding role of women in establishing and promoting EBF and MiS, providing good results in preventing malnutrition and improving many deficiencies. In addition, CHWs implementing interventions have proven to contribute to a better acceptance of the intervention among populations, consequently achieving better results in nutritional status. This review revealed that dietary diversity in pregnant women is crucial to reduce the prevalence of food-insecure households, leading to a greater growth of children; these may also be improved by educational programs.

Past studies have been insufficient to put an end to this burden of malnutrition; nevertheless, the results in this review are highly valuable to guide further actions. Future research needs to focus on the mechanisms underlying the observed association of educational programs and diet diversity. Further longitudinal studies that could explain poor child health as the result of scarce food-crop diversity would be particularly beneficial. It is urgent that the 2030 Agenda of Sustainable Development Goals and funding from global institutions are committed to implement the most effective combination of FS and WASH in those countries most affected by climate and health emergencies.

## Figures and Tables

**Figure 1 ijerph-18-04799-f001:**
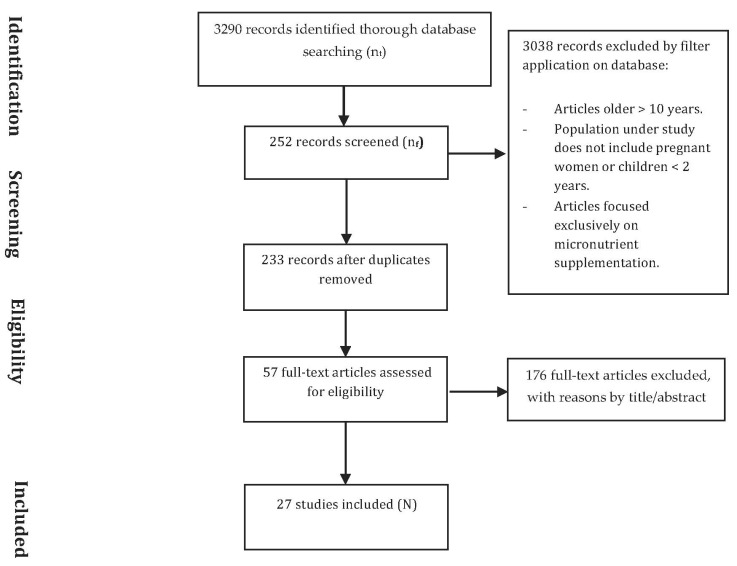
Flow chart showing the electronic research, identification, and selection process for the reviewed articles.

**Table 1 ijerph-18-04799-t001:** Search strategy and the number of articles (n_t_) found and selected (n_f_).

Keywords Searched by Boolean Operators	n_t_	n_f_
(food security) AND (nutritional status) AND (low and middle income countries)	41	2
((water OR sanitation OR hygiene) OR (WASH)) AND (nutritional status) AND (low and middle income countries)	58	1
(food security) AND (malnutrition) AND (low and middle income countries)	38	1
(pregnancy OR pregnant) AND ((water AND sanitation) OR WASH) AND (nutritional status OR malnutrition)	58	3
((water AND hygiene AND sanitation) OR WASH) AND (disease) AND (malnutrition OR undernutrition) AND ((pregnant or pregnancy) OR (children))	55	1
((water AND sanitation AND hygiene) OR WASH) AND (nutritional status)	155	12
(food security) AND (nutritional status)	2466	137
((water AND sanitation AND hygiene) OR (WASH)) AND (food security)	222	10
(water contaminants) AND (low and middle income countries)	8	1
(WASH) AND (intervention) AND (low and middle income countries)	38	2
(food security) AND (nutritional status) AND (low and middle income countries)	32	20
(water AND sanitation AND hygiene) AND (malnutrition OR undernutrition) AND (low and middle income countries)	8	6
(water AND sanitation AND hygiene) AND (food security) AND (low and middle income countries)	2	1
(pregnant OR pregnancy) AND (malnutrition OR undernutrition) AND (low and middle income countries)	109	55
	3290	252

**Table 2 ijerph-18-04799-t002:** Summary of general characteristics of selected articles, featured by type of intervention and classified considering the dimension approach and type of participant (**PLW**: Pregnant and lactating women; **WM**: Women mothers; **CH 6**: Children < 6 months; **CH 6–18**: Children 6–18 months; **CH 18**: Children > 18 months; **CHL5**: Children < 5 years; **CHW**: Community health workers; **WRA**: Women of reproductive age) in the intervention. Study design: cRCTC—Cluster Randomized Controlled Trial; RCT—Randomized Controlled Trial. The dimension of intervention has been marked with the symbol * (EduNC: Education and Nutrition Counselling; RUTF: Ready to Use Therapeutic Food; MiS: Micronutrient Supplement; MaS: Macronutrient Supplement and VO: Vouchers)

Type of Intervention	Author	Study Design	Study Location	Sample Size	Participant	Dimension of Intervention					Main Findings
						EduNC	RUTF	MiS	MaS	VO	
Food-based and nutritional	Kureishy S	cRCTC	Pakistan	7360	PLW			*	*		Effectiveness of food-based interventions in managing stunting
	Kimani-Murage EW	cRCTC	Kenya	1110 mother-child pairs	PLW	*					Basic nutritional training and/or provision of information may be adequate in improving exclusive breastfeeding rates in communities significantly
	Ara G	Study protocol	Bangladesh	205 mother-child pairs	WM and CH 6–18	*					The results provide robust evidence to improve the linear growth of children in developing countries by integrated intervention
	Frongillo EA	cRCTC	Bangladesh	300 pregnant women and 1000 recently delivered women	PLW	*					Household food insecurity was reduced in areas where nutrition-focused antenatal care was implemented. The integration of nutrition interventions into the maternal, newborn, child program was feasible and well implemented
	Menasria L	cRCTC	Cambodia	360	CHL5	*		*			Adding supplementary foods to education and counselling (CEN) activities did not improve young child nutritional status, as compared to CEN alone. Nutrition education and couselling alone was as effective as combining it with food supplements with regard to the impact on child anthropometry.
	Sigh S	Prospective RCT	Cambodia	121	CHL5		*				A locally produced ready-to-use therapeutic food (RUTF) might be as effective in terms of weight gain as an imported milk-based RUTF.
	Borg B	ProspectivecRCTC	Cambodia	540	CH 6–18		*	*	*		There is a need to develop locally produced and culturally acceptable RUSF, and to compare these with existing options for preventing malnutrition. This trial contributed to compare the effectiveness of supplementary foods with animal-source food and milk
	Le port A	cRCTC	Senegal	321	CH 18			*		*	Anaemia prevalence was very high at baseline (80%) and dropped to close to 60% at endline. Haemoglobin increased by 0.55 g/dL, 95%OI more in the intervention compared to the control group after one year, in models that controlled for potentially confounding factors. The impact was greater for boys compared to girls.
	Campbell RK	cRCTC	Bangladesh	5499	CH 6–18	*		*			Child dietary diversity is low. The meeting of the minimum dietary diversity (MDD) was equal or greater in the supplemented group with home foods than the control group at all ages. High socioeconomic status and maternal education were associated with increased odds of MDD, whereas household security was not associated with MDD.
	Na M	cRCTC	Bangladesh	14,600	WRA	*		*			Poverty and poor maternal education were recognized as important determinants that influence quality of diet. Economic and social strategies may be required to overcome food insecurity beyond maternal education itself.
	Fenn B	cRCT	Pakistan		CHL5					*	Different modalities of cash-based transfer work best to reduce the risk of wasting during a season where food insecurity is at its highest.
	Shaheen R	RCT	Bangladesh	4436	PLW			*			The combination of an early intervention to prenatal food supplementation and multiple micronutrient supplementation lowered mortality in children younger than five years and reduced the gap in child survival chances between social groups.
	Nguyen PH	RCT	Vietnam	4983	WRA and PLW			*	*		Poor dietary intakes and diet quality among underprivileged women (lower education, farers, and those living in households with lower socio-economic status) suggests that nutrition programs and policies should be linked with social development programs.
	Ryckman T	Difference-in- quasi-experimental approach	33 Countries in Sub-Saharan Africa	883,309	CHL5	*					Feed the Future’s activities were linked to notable improvements in stunting and underweight levels and moderate improvements in wasting in children <5 years.
	Ghosh-Jerath S	Cross-sectional study design	India	280 households per tribal group (3)	All	*					This study provides the first comprehensive examination of the food system of tribal communities. Interventions help to support the sustainable production and consumption of indigenous foods. Also, interventions address the burden of malnutrition in the tribal communities.
	Stevens B	Cross-sectional study	Bangladesh	288	PLW	*					Dietary diversity and household food security were sensitive to seasonal variations. Women had higher dietary diversity in autumn and winter. Household food insecurity peaked during autumn and was lesser in spring.
	Mutisya M	Prospective longitudinal trial	Kenya	6858	CHL5	*					Stunting was highest among poorest households and lowest among the least poor households. The proportion of children who were stunted was higher among mothers with lower levels of education.
	Lander RL	RCT	Guatemala, India, Pakistan and DR of Congo	988	PLW			*	*		Dietary patterns varied widely among sites. Significantly higher intakes of most key nutrients were observed in participants with adequate dietary diversity. There is a likely need for micronutrient supplementation in pregnancy as well as supporting the value of increasing dietary diversity.
	Hasan MT	Series of cross-sectional nationally representative DHS data	Bangladesh	28,941	CHL5	*					Children of mothers with secondary or higher education had a lower risk of childhood stunting, underweight and wasting. Promoting women’s education at least up to secondary level has great importance as a means to tackle the malnutrition in Bangladesh.
Food-based & WASH	Shafique S	Cluster randomized trial	Bangladesh	467	CH 6	*		*			The use of mineral- and vitamin-enhanced micronutrient powders reduced stunting significantly. On the contrary, the use of a water-based hygiene and sanitation did not have an additive effect.
	Mostafa I	Cross-sectional study	Bangladesh	370	WM	*					A total of 83% of the selected households experienced food insecurity. Mothers or caregivers in slums are less likely to have higher levels of education on hygienic practices. The poor personal hygiene practices of mothers, especially hand washing practices before food preparation, may be a contributing factor to the high contamination rate. All water samples were contaminated with faeces.
	Stewart CP	2 cRCTC	Kenya and Bangladesh	699 Kenya and 1470 Bangladesh	PLW, CHL5	*			*		The distribution of lipid-based nutrition supplements resulted in a lower prevalence of anaemia and iron deficiency in both countries. There were also reductions in the prevalence of low vitamin B12 status.
WASH	Schlegelmilch MP	Cluster randomized comparison study	Kenya	250	All						Improvements from baseline were observed, yet overall levels of latrine coverage are still low. The findings challenge the assumption that providing WASH infrastructure and education will result in behaviour change.
	Tsuka Y	Cross-sectional study	Indonesia	228 pairs of children and their caretakers	CHL5	*					Not using a towel for hand washing practices was significantly associated with an increased risk of stunting. Children from households using tap water as drinking water were associated with an increased risk of stunting and thinness compared with households using tank water. Children from households using open containers for water storage were associated with an increased risk of diarrhoea.
	Nurul TM	Cross sectional survey	Bangladesh	1000 households for observation, 1700 for cross sectional and 1000 for diarrhoea	CHW and CHL5	*					After 18 months of promoting key behaviours related to sanitation, hygiene and safe water, the improvements were modest. No reduction in the prevalence of diarrhoea or respiratory disease in children <5 years was observed. Eighteen months of the program were not sufficient to produce the targeted behaviour.
	Aw JYH	cRCTC	Timor-Leste	24 communities	PLW, WM, WRA, CHL5	*					No impact was found from community-based WASH intervention in terms of reducing *G. duodenalis* prevalence. Risk factors for *G. duodenalis* included living in a household with a child <5 years, living in a household with more than six people, and sampling during the rainy season.
	Lin A	cRCTC	Bangladesh	4102 available women, 6694 children	PLW	*					Individual hand washing and hygienic sanitation interventions reduced childhood Giardia infections. There were no effects from chlorinated drinking water and nutrition improvements. Combined WASH interventions provided no additional benefit in this context.

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
