# Peer review of "Interventions on Food Security and Water Uses for Improving Nutritional Status of Pregnant Women and Children Younger Than Five Years in Low-Middle Income Countries: A Systematic Review"

_ijerph, 2021, doi:10.3390/ijerph18094799_

Round 1
Reviewer 1 Report
Comments to author(s):
Paper review “Interventions on food security and water uses for improving nutritional status of pregnant women and children below five years in low-middle income countries: A systematic review”.
It is an interesting paper and presents a good reading.
Concerns raised include:
- Abstract: In the abstract, lines 14-16, the methods of the study with more specific information would be helpful.
- Introduction: There is a lack of integrated academic literature support. There are no references for some of the statements. e.g. lines 47-53.
- Materials and Methods: Lines 94-95, add a brief description of ‘standard methods’ and include reference(s).
- Results: Line 242, correct the style of the references. Lines 259-265, reference(s) needed.
- Other comments: Please read the paper carefully with regard to correct English.
Author Response
Manuscript ID: ijerph-1186502
Title: Interventions on food security and water uses for improving nutritional status of pregnant women and children below five years in low-middle income countries: A systematic review
Food Bioprocess and Technology
We appreciate the positive and interesting comments from reviewers to enhance the quality of our manuscript. Corrections have been highlighted in blue along the text to facilitate further review. An itemized list of changes as well as our comments is presented below.
Response to Reviewer 1 Comments
- Abstract: In the abstract, lines 14-16, the methods of the study with more specific information would be helpful.
Response 1
Following your suggestions the description of the methodology was detailed as: Using PRISMA procedures, a systematic review was conducted by searching in biomedical databases clinical trials and interventions for children and pregnant women.
- Introduction: There is a lack of integrated academic literature support. There are no references for some of the statements. e.g. lines 47-53.
Response 2
The references for lines 47-53 have been checked and adapted to your suggestions
- Materials and Methods: Lines 94-95, add a brief description of ‘standard methods’ and include reference(s).
Response 3
The standand methods were specified as: we conducted a systematic literature search using combination of keywords as described in PRISMA methodology.
Furthermore the reference of PRISMA methodology was included.
- Results: Line 242, correct the style of the references. Lines 259-265, reference(s) needed.
Response 4
The writing references have been checked
- Other comments: Please read the paper carefully with regard to correct English.
Response 5
The English grammar on the whole manuscript have been reviewed

Reviewer 2 Report
While this is a very interesting systematic literature review, there is a lack of methodological development which the authors should improve prior to its potential publication. In particular:
- the authors should provide a justification to support the use of this methodology, and the rationale of databases and keywords, period of time, publications that were not eligible and language.
- the authors should develop the eligibility assessment and how it is possible to replicate the notion of ‘relevant to qualitative synthesis’ that resulted in the analysis of 27 articles.
- conclusion should explain the limitations of the research and expand the opportunities for further research.
Round 2
Reviewer 2 Report
While the authors have revised the paper, they do not provide a justification to support the use of databases and the conclusion is not enough developed according to previous comments.
